# Determination of Market, Field Samples, and Dietary Risk Assessment of Chlorfenapyr and Tralopyril in 16 Crops

**DOI:** 10.3390/foods11091246

**Published:** 2022-04-26

**Authors:** Hong Li, Fengshou Sun, Shuai Hu, Qi Sun, Nan Zou, Beixing Li, Wei Mu, Jin Lin

**Affiliations:** 1Shandong Provincial Key Laboratory for Biology of Vegetable Diseases and Insect Pests, College of Plant Protection, Shandong Agricultural University, Taian 271018, China; 2018110350@sdau.edu.cn (H.L.); H15564801860@163.com (S.H.); zounan1226@163.com (N.Z.); libeixing@126.com (B.L.); muwei@sdau.edu.cn (W.M.); 2Research Center of Pesticide Environmental Toxicology, Shandong Agricultural University, Taian 271018, China; sfsun1990@163.com (F.S.); sunqi05172022@163.com (Q.S.)

**Keywords:** chlorfenapyr and tralopyril, market monitoring, cabbage, dissipation and final residues, dietary risk assessment

## Abstract

The frequent and massive use of chlorfenapyr has led to pesticide residues in crops, threatening food safety and human health. However, there is limited research on the detection of tralopyril, which is the major metabolite of chlorfenapyr with high toxicity. This study aimed to develop a novel, sensitive, and highly efficient method for the determination of chlorfenapyr and tralopyril residues in 16 crops. The optimized purification procedure provided satisfactory recovery of 76.6–110%, with relative standard deviations of 1.3–11.1%. The quantification values of pesticides in crop matrixes were all 0.01 μg kg^−1^. The optimal method was adopted to determine the chlorfenapyr and tralopyril residues in field trials in 12 regions in China and monitor their residues in 16 agricultural products. The results of the dissipation and terminal residue experiments show that the final residue of chlorfenapyr was less than MRL (maximum residue limit) and no tralopyril was detected in the field samples. Moreover, the qualification proportion of these residues in market samples were up to 99.5%. The RQ (risk quotient) values of chlorfenapyr and chlorfenapyr with consideration of tralopyril were both apparently lower than an RQ of 100%, indicating an acceptable level. This research provides a thorough long-term dietary risk evaluation on chlorfenapyr and tralopyril and would provide reference for their scientific and safe utilization.

## 1. Introduction

Chlorfenapyr, a new pyrrole insecticide, is widely used in different agricultural products, such as Chinese cabbage, cabbage, citrus, apple, tea, and other vegetables, fruits, and other crops, due to their high-efficiency and broad-spectrum characteristics. Chlorfenapyr has excellent pest control against a number of species, such as *Liriomyza* sp., *Frankliniella occidentalis*, *Spodoptera exigua,* and *Spodoptera litura* [1]. Chlorfenapyr is a highly concentrated pesticide, which can be toxic to humans, birds, fish, and silkworms, and can also damage the DNA of liver, spleen, and kidney cells of mice [2,3]. In the cultivation process, vegetables are vulnerable to pests and diseases, especially under open field conditions [4,5]. In particular, cabbage is an important Florida fresh-market crop with a 2021 production value of $45.34 million [6]. Moreover, the China Pesticide Information Network reported cabbage with the largest number of registrations of chlorfenapyr, accounting for 70%, and the leaves are wrapped in layers, making them prone to pesticide residues and accumulation [7]. Pesticides employed to ensure the high yield of food and to minimize post-harvest losses have brought with them negative health risks [1,8]. Therefore, more attention should be paid to pesticide residues and food safety.

According to the JMPR (Joint Food and Agriculture Organization of the United Nations and World Health Organization Meeting) report in 2018, the residue and dietary risk assessment of chlorfenapyr in plants were defined as the sum of parent plus 10-fold major metabolite (tralopyril) with high toxicity. Therefore, it is important to conduct a simultaneous investigation on its metabolite tralopyril for dietary risk assessment. At present, previous research has reported the detection of chlorfenapyr in vegetables [9,10,11,12], fruit [13,14,15], tea [9,16], and cereals [1,17] using various methods, mainly including liquid chromatography (LC), liquid chromatography with tandem mass spectrometry (LC-MS/MS), gas chromatography, or gas chromatography with tandem mass spectrometry (GC-MS/MS). Some research found that chlorfenapyr suspension concentrate formulation degraded in cabbage and celery, with half-life values of 3.9 days and 6.3 days, respectively [18,19]. At least a 15-day safe waiting period was recommended before harvesting grape berries once chlorfenapyr was applied at 144 g ha^−1^ [20].

However, to our knowledge, there have been few reports focused on the residues of tralopyril on crops. Furthermore, research on the market sample monitoring of registered crops, terminal residue analysis and dissipation behaviour for cabbage in field trials, and dietary risk assessments of chlorfenapyr and its metabolites (tralopyril) are still scarce. It is essential to judge the persistence of this compound as well as any impacts on consumer safety associated with its use. A dietary risk assessment of chlorfenapyr and tralopyril residues in 16 crops will provide a scientific basis for adequate supervision, safe production, consumption guidance, and maximum residue level (MRL) issuance and revision. This study could provide a significant reference for establishing standards for the secure and rational use of chlorfenapyr, formulating MRLs, monitoring the quality safety of agri-food, and protecting consumer health.

Therefore, this study was designed to determine chlorfenapyr and tralopyril residues in registered crops in China due to food safety concerns. This study aimed to (1) develop and validate a sensitive and straightforward method to detect and quantify chlorfenapyr and tralopyril residues in 16 crops; (2) monitor the residue status of chlorfenapyr and tralopyril on 16 kinds of agricultural products collected from Tai’an, Shandong province; (3) investigate the terminal residues and dissipation dynamics on cabbage in 12 regions in China; and (4) assess a long-term dietary risk for Chinese adults.

## 2. Materials and Methods

### 2.1. Chemicals and Reagents

Reference standards of chlorfenapyr (98.1%) and tralopyril (99.3%) were purchased from Dr. Ehrenstorfer (Germany Dr. Ehrenstorf GmbH, Augsburg, Germany). Ultra-high-performance liquid chromatography (UHPLC)-grade acetonitrile and dichloromethane were purchased from Merck (Sigma-Aldrich, Darmstadt, Germany). Analytical grade sodium chloride (NaCl) and anhydrous magnesium sulfate (MgSO_4_) were purchased from Sinopharm (Thermo scientific, Shanghai, China). The purification agents of GCB, PSA, and C_18_ were purchased from Agela Technologies Co., Ltd. (Agela, Tianjin, China). The 240 g L^−1^ chlorfenapyr SC (suspension concentrate) was purchased from Shandong United Pesticide Industry Co., Ltd. (Rennes, Jinan, China).

The stock solutions (1000 mg L^−1^) of chlorfenapyr and tralopyril were prepared by diluting dichloromethane and acetonitrile, respectively. Two kinds of stock solutions were serially diluted to obtain working standard solutions at 0.001, 0.005, 0.01, 0.05, 0.1, and 0.5 mg L^−1^ concentrations. All solutions were stored at 4 °C in the dark.

### 2.2. Sample Collection

(1).The collection of market samples

There were 20 points to collect samples from a total of eight large supermarkets, five small supermarkets, and seven farmer’s markets of Tai’an, Shandong province, shown in Appendix A. Each collection point contained 16 registered crops using chlorfenapyr: Chinese cabbage, beans, cabbage, cucumber, hairy gourd, eggplant, cowpea, Welsh onion, cabbage mustard, leek, pak choi, citrus, pear, apple, tea, and ginger. Sixteen kinds of agricultural products numbered in sequence were collected, making a total of 320 samples. The amount of each sample should not be less than 500 g, of which the tea sample should not be less than 100 g. Samples were stored at −20 °C.

(2).Field trials

The field trials contained residual dissipation and terminal residue experiments. Terminal residue experiments were carried out in 12 provinces, including Liaoning, Shanxi, Beijing, Shandong, Shanghai, Anhui, Hunan, Jiangxi, Guangxi, Chongqing, Guizhou, and Guangdong, and residual dissipation experiments were carried out in 4 provinces, including Beijing, Shanghai, Hunan, and Guangdong. Field test sites, crop varieties, and test types are shown in Appendix A.

Given the heavy workload of crop residual tests, the present research could not carry out a residual test on all 16 registered crops. Therefore, as the most common registered crop using chlorfenapyr, accounting for 70% of the registered products, cabbage was selected as the representative for residual tests.

To investigate the terminal and dissipation residue levels of chlorfenapyr and tralopyril, the 240 g L^−1^ chlorfenapyr SC was diluted with water and sprayed at a dose level of 120 g active ingredient per ha (g ha^−1^) (registered high dosage in cabbage). The residual experimental plots in the peak development stage of young larvae of cabbage diamondback moth were sprayed with a power sprayer for twice times with an interval of 7 days. The untreated plots with the same size but no chlorfenapyr application were simultaneously compared. Each treatment had three replicate plots and one control plot (no chlorfenapyr, water only) with an area of 50 m^2^. Cabbage samples (2 kg) of terminal residue experiments were randomly collected at 14 d and 21 d after the last application. Furthermore, representative samples (2 kg) for dissipation analysis in cabbage were randomly collected from each plot at 2 h, 3 d, 7 d, 14 d, and 21 d after the last application. Three parallel samples were collected each time. Blank control was randomly collected from untreated trial plots before pesticide application. Finally, all samples were maintained at −20 °C before further analysis.

### 2.3. Sample Extraction and Purification

All samples were prepared following a QuEChERS-based method [21]. Each sample was ground in a blender. For vegetable and fruit samples, 10 g of the homogenized sample was taken in a 50 mL centrifuge tube and mixed with 20 mL of acetonitrile. Amounts of 5 g of grain and oil samples and 2.5 g of tea samples were weighed for homogenization. Then, 10 mL of water was added for 30 min, standing for 30 min. Subsequently, 20 mL of acetonitrile was added to the mixture and shaken for 3 min, after which NaCl (1 g) and MgSO_4_ (4 g) were added to the tube and shaken vigorously for 5 min, followed by centrifugation at 1425 g for 5 min. Then, 1.5 mL of the extracted solution was transferred to a 2 mL centrifuge tube equipped with different sorbent mixtures (PSA/C_18_/ GCB and 150 mg MgSO_4_). The details of the different combinations of purifiers used to obtain the optimal purification method are shown in Appendix A. After shaking (5 min) in a mechanical shaker and centrifugation (8910× *g* for 2 min), the supernatant of chlorfenapyr was blow-dried by nitrogen, which was redissolved with dichloromethane. Finally, the redissolved solution of chlorfenapyr and the supernatant after purification of tralopyril were filtered through a 0.22 μm membrane filter for GC-MS/MS and UHPLC-MS/MS analysis, respectively.

### 2.4. Instrumental Parameters

#### 2.4.1. UHPLC-MS/MS Parameters

UHPLC-MS/MS analysis of tralopyril was performed using a Waters ACQUITY UPLC^®^BEH C_18_ (50 mm × 2.1 mm; i.d., 1.7 μm) equipped with an electrospray ionization (ESI) source. The injection volume was 1 μL, and the flow rate was 0.30 mL min^−1^. Deionized water containing 0.05% formic acid water (mobile phase A) and methanol (mobile phase B) were used for the gradient elution program. The detailed parameters are listed in Appendix A. The UHPLC conditions were optimized to obtain a fast and reliable separation of tralopyril. Multiple reaction monitoring (MRM) was utilized to selectively detect and quantify pesticides in multiple crops. The MS/MS was carried out using an electrospray ionization source set in negative ion mode.

Mass spectrometry was carried out in multiple reaction monitoring (MRM) mode and negative ionization mode (ESI^−^). The conditions adopted were as follows: capillary voltage of 0.5 kV, desolventization temperature of 350 °C, and ion transfer tube temperature of 325 °C. Detailed conditions are shown in Appendix A.

#### 2.4.2. GC-MS/MS Parameters

GC-MS/MS analysis of chlorfenapyr was performed using an EVO TSQ8000 TG-5MS (15 mm × 0.25 mm; i.d., 0.25 μm) equipped with an electron bombardment ion source (EI). The injection volume was 1 μL with a flow rate of 1.2 mL min^−1^. Nitrogen (99.99%) was used as a carrier gas. The injector temperature was 280 °C with the splitless mode. The initial temperature was 100 °C for 1 min and then raised to 280 °C for 5 min at a speed of 30 °C min^−1^. The GC conditions were optimized to obtain a fast and reliable separation of chlorfenapyr. Selective reaction monitoring (SRM) was utilized to selectively detect and quantify pesticides in multiple crops. The MS/MS was carried out using an electron bombardment ion source (EI). The detector temperature was programmed at 300 °C. The MS parameters were: acquisition mode, selective reaction monitoring (SRM); ion source temperature, 300 °C; ionization voltage, 70 eV. The detailed parameters were listed in Appendix A.

### 2.5. Method Validation

According to the guidance document on method validation and quality control procedures for the analysis of pesticide residues in food and feed [22], the proposed method was verified for linearity, matrix effect (ME), accuracy, precision, limit of detection (LOD), and limit of quantification (LOQ). Linearity of the method was verified using blank matrix standards at six different concentrations (0.001, 0.005, 0.01, 0.05, 0.1, and 0.5 mg L^−1^) for chlorfenapyr and tralopyril. The ME is calculated as slopes of the analytical curves obtained from solutions prepared in the solvent and those in the blank matrix extract [23]. The recovery rate and relative standard deviation (RSD, %) of the method using different combinations of purification agents were determined using representative crops of different types (vegetable and fruit samples: cabbage and apple; grain and oil samples: wheat and peanut; tea), spiked to a concentration of 0.01 mg kg^−1^. Furthermore, the accuracy and precision of the method, with the optimal combination of purifying agents identified based on the recovery rate and RSD, were assessed using 16 kinds of blank samples registered with chlorfenapyr and spiked to 0.01, 1, and 10 mg kg^−1^ concentrations of chlorfenapyr and tralopyril (except tea blank samples spiked to 0.01, 1, and 20 mg kg^−1^), with five replicates each. A total of 16 kinds of registered crops using chlorfenapyr included Chinese cabbage, beans, cabbage, cucumber, hairy gourd, eggplant, cowpea, Welsh onion, cabbage mustard, leek, pak choi, citrus, pear, apple, tea, and ginger. Sensitivity was evaluated by determining the LOD and LOQ. The LOQ of the method was defined as the lowest spiked concentrations that met the criteria. The LOD was defined as the lowest standard curve [24].

### 2.6. Dietary Risk Assessment

The long-term dietary risk assessment of pesticide residues in five types of foods (dark vegetables, light vegetables, fruits, salt, and soy sauce) for an average amount of Chinese adults were calculated using Equations (1) and (2) [25,26]:NEDI = STMR_i_ × F_i_(1)
RQ (%) = NEDI/ADI × 100%(2)
where NEDI refers to the assessed daily intake (mg kg^−1^ day^−1^), STMR_i_ (supervised trials median residue level, mg kg^−1^) represents the median concentration of pesticide residues in tomatoes of China, F_i_ (kg day^−1^) is the average daily intake of special farm food in China, ADI (0.03 mg kg^−1^ day^−1^) is the acceptable daily intake of pesticide residues set by GB-2763-2019 China, and is the average body weight of Chinese adults of 63 kg [27].

The RQ represents the risk quotient. When the RQ ≤ 100%, it indicates acceptable risk, otherwise it did not [28,29,30].

## 3. Results and Discussion

### 3.1. Optimization of UHPLC-MS/MS and GC-MS/MS

#### HPLC–MS/MS Optimization Conditions

The precursor ions were fragmented into two intense fragment ions under the MS conditions, chosen as the quantitative and qualitative ions. The conditions adopted were as follows: the optimized parameters of the precursor ion, retention time, and product ion, as summarized in Appendix A. The precursors to tralopyril ion transitions were performed for MRM scans with the following product transitions: *m/z* 349.0→131.0 and 349.0→81.0. The UHPLC-MS/MS chromatogram of tralopyril is shown in Appendix A.

The precursor ions were fragmented into two intense fragment ions under the MS conditions, chosen as the quantitative and qualitative ions. The conditions adopted were as follows: the optimized parameters of the precursor ion, retention time, and product ion, as summarized in Appendix A. The precursors to chlorfenapyr ion transitions were performed for SRM scans with the following product transitions: *m/z* 59.1→31.1 and 59.1→27.1. The GC-MS/MS chromatogram of chlorfenapyr was shown in Appendix A.

### 3.2. Optimization of Sample Pretreatment

Various crops are rich in vitamins, carotenes, trace elements, proteins, sugars, organic acids, fat, and cellulose. These diverse impurities make the sample matrix highly complex for analysis. Therefore, for different types of crops, it is necessary to achieve a good purification effect using various combinations of purification materials before analysis. Currently, PSA, C_18_, and GCB are most commonly used to adsorb impurities. The PSA sorbent has a strong adsorption capacity for metal ions, fatty acids, sugars, and fat-soluble pigments but has a poor purification efficiency on pigments in tomatoes [31]. C_18_ is commonly used in the QuEChERS method owing to its strong adsorption of nonpolar impurities such as fats, sterols, and volatile oils [32]. Graphitized carbon black (GCB) is available for removing color pigments [33].

In the current study, different purification materials were combined to propose the best purification strategy in the pretreatment of representative crops (vegetable and fruit samples: cabbage and apple; grain and oil samples: wheat and peanut; tea) for pesticide residue analysis. The average recovery was used to assess the purification effect, and values ranging from 70% to 120% were considered excellent [34]. As displayed in Figure 1A,B, the average recovery of chlorfenapyr (and tralopyril in the cabbage and apple matrixes, using different amount combinations of PSA + C_18_ (10 mg + 40 mg, 20 mg + 30 mg, 40 mg + 10 mg)), ranged from 55.8%–68.3%, 59.3–67.9%, and 92.1%–100.2%, respectively, with RSDs of 2.6%–6.3%. The average recovery of chlorfenapyr and tralopyril in the wheat and peanut matrixes using the different amount combinations of PSA + C_18_ (10 mg + 40 mg, 20 mg + 30 mg, 40 mg + 10 mg), ranged from 84.1%–100.6%, 57.4–68.4%, and 120.9%–128.9%, respectively, with RSDs of 3.3%–6.7%.

Furthermore, the average recovery of chlorfenapyr and tralopyril in the tea matrixes using the different amount combinations of PSA + C_18_ + GCB (10 mg + 40 mg + 5 mg, 20 mg + 30 mg + 5 mg, 40 mg + 10 mg + 5 mg), ranged from 129.7%–131.4%, 55.1–62.7%, and 98.9%–101.2%, respectively, with RSDs of 2.6%–6.2%. The results indicate PSA as having a good purifying effect on medium-strong polar impurities and can be selected as the adsorbent for vegetables, fruits, and tea, due to vegetables and fruits containing many medium-strong polar impurities such as sugar, vitamins, and pigment. Wheat and peanut contain weaker polar impurities such as fat, protein, phenols, and polysaccharides, which can be well-purified by C_18_. Therefore, C_18_ is recommended as the main adsorbent for purifying grain and oil substrates (wheat and peanut). Moreover, for samples with higher pigment content such as tea, a certain amount of GCB is needed for pigment adsorption. In conclusion, the purification strategies of (1) vegetables and fruits, (2) grain and oil substrates, and (3) tea were determined for the various purification combinations: ((1) 40 mg PSA + 10 mg C_18_ + 150 mg MgSO_4_, (2) 10 mg PSA + 40 mg C_18_ + 150 mg MgSO_4_, and (3) 40 mg PSA + 10 mg C_18_ + 5 mg GCB + 150 mg MgSO_4_).

### 3.3. Method Validation

The linearity, precision, matrix effect, reproducibility, and sensitivity were demonstrated to validate the developed method in this study. As shown in Table 1, the calibration curve of chlorfenapyr and tralopyril show an optimal linear relationship with a coefficient of determination (*R*^2^) ≥ 0.9966 within the tested concentration range of 0.001–0.5 mg L^−1^. The matrix effect (ME), defined as ion enhancement or suppression, was evaluated by comparative analyses using two kinds of samples based on a previously proposed approach [35]. The MEs of chlorfenapyr and tralopyril in 16 crops ranged from 0.80 to 1.02, indicating no obvious matrix effect [36]. The LOQs of chlorfenapyr and tralopyril in crops were all 0.01 μg kg^−1^. Furthermore, as shown in Figure 2 and Table 1, the recovery tests of chlorfenapyr and tralopyril in vegetable and fruit samples were performed using the optimal purifier combination PSA + C_18_ + MgSO_4_ (40 mg + 10 mg + 150 mg), with a range from 76.6% to 105.3% and RSDs of 1.5%–11.1% at three spiking levels (0.01, 0.1, and 10 mg kg^−1^). The recovery tests of chlorfenapyr and tralopyril in grain and oil samples were performed using the optimal purifier combination PSA + C_18_ + MgSO_4_ (10 mg + 40 mg + 150 mg), with a range from 85.4% to 110.0% and RSDs of 2.5%–6.5% at three spiking levels (0.01, 0.1, and 10 mg kg^−1^). The recovery tests of chlorfenapyr and tralopyril in tea were performed using the optimal purifier combination PSA + C_18_ + GCB + MgSO_4_ (40 mg + 10 mg + 5 mg + 150 mg), with a range from 83.5% to 101.2% and RSDs of 1.7–6.2% at three spiking levels (0.01, 0.1, and 20 mg kg^−1^). These findings indicate that the method has acceptable accuracy and precision. Therefore, the proposed method can be used to quantify pesticide residues in multiple crop samples. This method was suitable for the detection of five types of crops (vegetable, fruit, grain, oil, and tea substrates), which shows wide applicability. Additionally, the method can improve the detection efficiency and reduce the use of organic reagents by omitting the rotary steaming step. Moreover, the method is friendly to the environment and reduces costs. In conclusion, this method has the advantages of high sensitivity, high accuracy, high stability, acceptable economy, and wide application.

### 3.4. Residues of Chlorfenapyr and Tralopyril in 16 Market Samples

The collected samples were determined according to the proposed method. They were regarded as detected with the residual amount of LOQ over 0.01 mg kg^−1^. The residues of chlorfenapyr and tralopyril in 16 crops collected from 20 cities (Tai’an, Shandong, China) are listed in Table 2. The results of Table 2 show that only 38 out of the total 320 samples contained pesticide residues, with a detection rate of 14.5%. The residue of one sample exceeded the standard with a rate of 0.31% (the residual level of 0.462 mg kg^−1^ in CjYP-I08 cabbage mustard), and the qualified rate was more than 99.5%. All in all, the detection rate of chlorfenapyr was more than 10%, but the over-standard rate was less than 0.5%, indicating that chlorfenapyr had a wide application and favorable security. Furthermore, only one sample of tralopyril was detected with a residual amount of 0.01 mg kg^−1^ in the CJYP-A07 cowpea, within the levels of the MRLs in China [37].

### 3.5. Terminal Residues and Dissipation Behaviors of Chlorfenapyr in Cabbage Samples

The dissipation behaviors of chlorfenapyr in cabbage from Beijing, Shanghai, Hunan, and Guangdong were analyzed. Figure 3 shows that the dissipation dynamics of chlorfenapyr in the four sites can be described as follows: Y = 1.1189e^−0.224x^ (Beijing, R = 0.9882, t_1/2_ = 3.09 days), Y = 0.5441e^−0.213x^ (Shanghai, R = 0.9904, t_1/2_ = 3.25 days), Y = 0.8429e^−0.323x^ (Hunan, R = 0.9919, t_1/2_ = 2.15 days), and Y = 2.3824e^−0.129x^ (Guangdong, R = 0.9838, t_1/2_ = 5.37 days). The dissipation results show that chlorfenapyr was obviously digested after application, but no tralopyril was detected during the whole test. Consequently, the proportion of tralopyril in the metabolites of chlorfenapyr was very low, and the formulation of chlorfenapyr contained no tralopyril. In addition, the degradation rate of chlorfenapyr was fast, which indicates that it is not likely to cause food safety problems due to low enrichment in crops. According to the available literature, the dissipation of compounds in crops can be influenced by their chemical properties and multiple environmental factors (volatilization, hydrolysis, wash off, and photodegradation) under field conditions [38,39].

The terminal residue of pesticides in cabbage in 12 regions are listed in Appendix A. The data showed that the terminal residues of various regions ranged from 0.046–0.210 mg kg^−1^ in Liaoning, 0.031–0.310 mg kg^−1^ in Shanxi, <0.01–0.067 mg kg^−1^ in Beijing, <0.01–0.070 mg kg^−1^ in Shandong, <0.01–0.016 mg kg^−1^ in Shanghai, 0.014–0.121 mg kg^−1^ in Anhui, <0.01 mg kg^−1^ in Hunan, <0.01–0.213 mg kg^−1^ in Jiangxi, <0.01–0.089 mg kg^−1^ in Guangxi, <0.01–0.337 mg kg^−1^ in Chongqing, <0.01–0.015 mg kg^−1^ in Guizhou, and 0.144–0.471 mg kg^−1^ in Guangdong, with a two times interval of 7 d post-application, according to the recommended dosage. Summarily, the final residual amount of chlorfenapyr in cabbage was less than 0.01–0.471 mg kg^−1^ for 16 regions, within the maximum residue limit (MRL) for chlorfenapyr on cabbage (1 mg kg^−1^) [37]. According to the formula mode of JMPR (2018), which was described in Section 2.6, the final residue is considered to be the residual sum of the conversion of tralopyril and chlorfenapyr. The final residual levels of the 12 sites observed in this study were arranged in ascending order, and the results in Table 3 show that the STMR (supervised trials median residue) and HR (the highest residue) were 0.105 mg kg^−1^ and 0.471 mg kg^−1^, respectively. The present research suggests that the 240 g L^−1^ chlorfenapyr (SC) can be securely used on cabbage farmlands at the recommended dosage. These results can be used to assess the dietary safety of cabbage consumption.

### 3.6. Dietary Risk Assessment

The values of STMR and market monitoring obtained in this study and the established MRL of chlorfenapyr in registered crops are summarized to assess dietary safety (Appendix A). The dietary risk assessment was calculated with and without tralopyril. The NEDI and RQ were calculated according to the dietary intake and weight survey data combined with the pesticide residues in five types of foods (dark vegetables, light vegetables, fruits, salt, and soy sauce). The results in Table 4 show that the NEDI values of chlorfenapyr and the sum of chlorfenapyr and tralopyril were 1.322 mg and 1.340 mg, respectively. The RQs of chlorfenapyr and the sum of chlorfenapyr and tralopyril in registered crops were 65.1% and 66.0%, respectively. The RQ results show no significant difference between the two, which indicates a low contribution of tralopyril to the dietary risk. We found the RQ values of chlorfenapyr and tralopyril that were less than 100% to be an acceptable level of pesticides detected in registered crops, indicating safe consumption [16]. The dietary risk assessment of chlorfenapyr is evaluated with and without tralopyril, because as the metabolite of chlorfenapyr, tralopyril has high toxicity in comparison to the parent chlorfenapyr and probably causes greater food safety problems. Both chlorfenapyr and tralopyril were assessed more comprehensively than in other studies, which only consider food risk of chlorfenapyr. Therefore, the present study can more valuably reflect food safety and health risks in humans [12,16,24,40,41,42]. However, with the increase of the scope of registration and application of chlorfenapyr in the future, the dietary risk would be increased. Furthermore, the RQ value of 66.0% is a high level, thus there is a need for constant attention to the dietary risk of chlorfenapyr in crops.

## 4. Conclusions

An effective method to quantify chlorfenapyr and tralopyril residues in 16 crops was developed, employing the QuEChERS procedure combined with UHPLC/GC-MS/MS. The purification methods were optimized using various combinations of purification materials. The optimal method was adopted to determine market samples, terminal residues, and dissipation behavior of chlorfenapyr and tralopyril in agricultural products. The results of the dissipation and terminal residue experiments show that the final residues of chlorfenapyr were less than MRL, and no tralopyril was detected in cabbage. Moreover, the qualification rate of pesticide residues in market samples were 99.5%. These data can provide effective instructions to properly use the pesticide and ensure food safety. The total RQ values of chlorfenapyr and chlorfenapyr with consideration of tralopyril on various crops were greatly lower than RQ = 100%, which indicate that the long-term dietary risk was correspondingly low for Chinese adults. This research could guide the rational use of chlorfenapyr and tralopyril and serve as a reference for the establishment of an MRL in China. Furthermore, conducting risk assessment of pesticide residues contributes to food safety risk management, risk communication, and consumer health.

## Figures and Tables

**Figure 1 foods-11-01246-f001:**
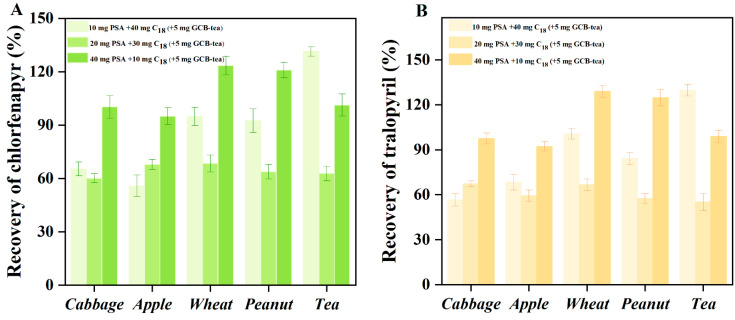
Recoveries of chlorfenapyr (**A**) and tralopyril (**B**) in 5 representative samples under different purification combinations. Note: Five replicates with each treatment. The spiked concentration of samples was 0.01 mg kg^−1^.

**Figure 2 foods-11-01246-f002:**
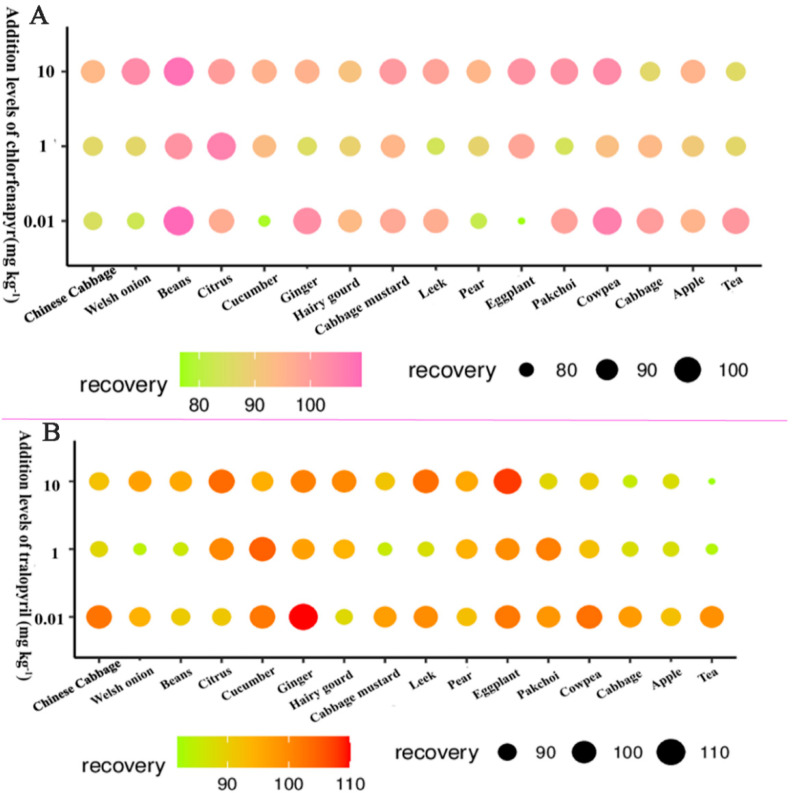
Recoveries of chlorfenapyr (**A**) and tralopyril (**B**) in multiple matrices. Note: the colors and circle sizes both show the amount of recovery.

**Figure 3 foods-11-01246-f003:**
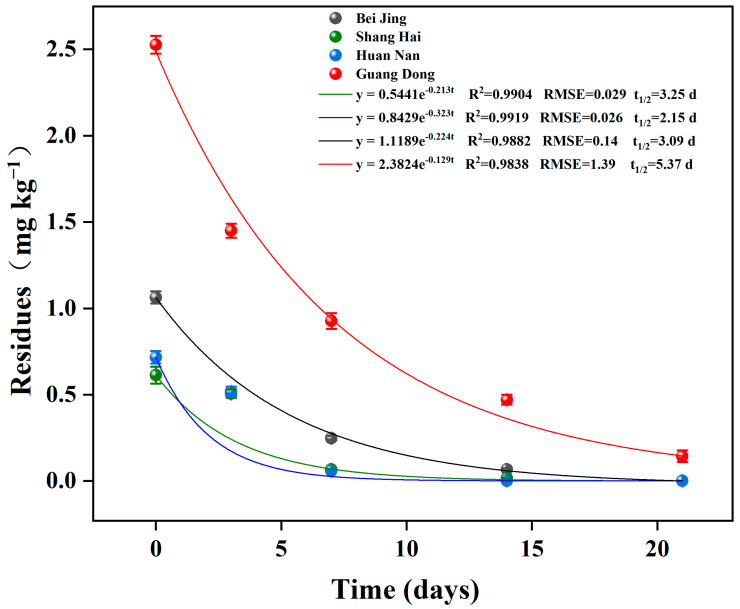
Dissipation behavior of chlorfenapyr in cabbage in four regions in China.

**Table 1 foods-11-01246-t001:** Validation of linearity, ME, and LOQ of chlorfenapyr and tralopyril.

Matrix	Pesticides	(y =) Standard Curve	*R^2^*	ME	LOQ (mg kg^−1^)	RSDs (*n* = 5, %)		
						Addition Levels (mg kg^−^^1^)		
						0.01	1	10 (20*)
Solvent	Chlorfenapyr	3,754,158 x − 22,467	0.9989	-	0.01	-	-	-
	Tralopyril	2,421,878 x + 13,065	0.9991	-	0.01	-	-	-
Chinese cabbage	Chlorfenapyr	3,660,723 x + 4630	0.9993	0.98	0.01	7.4	5.0	4.7
	Tralopyril	2,555,390 x + 16,483	0.9989	1.06	0.01	8.0	5.9	3.7
Welsh onion	Chlorfenapyr	3,310,947 x + 6823	0.9991	0.88	0.01	8.4	3.5	1.5
	Tralopyril	2,445,312 x + 22,932	0.9969	1.01	0.01	9.0	4.6	2.2
Beans	Chlorfenapyr	3,096,302 x + 3941	0.9994	0.82	0.01	7.1	3.1	4.6
	Tralopyril	2,529,608 x + 15,401	0.9984	1.04	0.01	10.3	6.5	2.9
Citrus	Chlorfenapyr	3,428,831 x − 3229	0.9993	0.91	0.01	7.1	3.2	3.1
	Tralopyril	2,305,545 x + 13,773	0.9993	0.95	0.01	5.4	5.6	1.5
Cucumber	Chlorfenapyr	3,507,036 x + 748	0.9996	0.93	0.01	5.2	5.7	3.4
	Tralopyril	2,562,382 x + 7001	0.9994	1.06	0.01	7.3	3.0	3.1
Ginger	Chlorfenapyr	3,575,404 x + 2848	0.9994	0.95	0.01	9.2	4.7	2.6
	Tralopyril	2,370,895 x + 10,212	0.9997	0.98	0.01	6.2	4.0	4.4
Hairy gourd	Chlorfenapyr	3,798,376 x + 619	0.9998	1.01	0.01	9.7	3.5	2.9
	Tralopyril	2,384,714 x + 17,640	0.9987	0.98	0.01	3.3	2.0	4.5
Cabbage mustard	Chlorfenapyr	3,317,870 x + 13,073	0.9993	0.88	0.01	10.0	3.1	4.0
	Tralopyril	2,567,317 x + 11,381	0.9992	1.06	0.01	7.1	4.1	1.7
Leek	Chlorfenapyr	34,89,748 x + 13,280	0.9996	0.93	0.01	11.1	4.6	3.5
	Tralopyril	2,500,523 x − 7439	0.9966	1.03	0.01	7.2	3.0	2.4
Pear	Chlorfenapyr	3,063,084 x + 18,341	0.9999	0.82	0.01	5.2	3.8	4.5
	Tralopyril	2,425,250 x + 12,353	0.9993	1.00	0.01	9.1	3.4	2.4
Eggplant	Chlorfenapyr	3,409,683 x + 6121	0.9996	0.91	0.01	9.7	2.7	2.4
	Tralopyril	2,221,298 x + 21,405	0.9976	0.92	0.01	6.1	4.8	3.3
Pak choi	Chlorfenapyr	3,275,217 x + 13,699	0.9991	0.87	0.01	7.8	4.2	1.3
	Tralopyril	2,379,877 x + 546	0.9998	0.98	0.01	7.4	3.4	1.4
Cowpea	Chlorfenapyr	3,015,524 x + 28,920	0.9994	0.80	0.01	6.9	5.6	2.5
	Tralopyril	2,555,844 x + 9531	0.9995	1.06	0.01	8.9	4.2	5.4
Cabbage	Chlorfenapyr	3,547,825 x − 5212	0.9991	0.95	0.01	6.3	6.2	2
	Tralopyril	2,458,083 x + 12,693	0.9992	1.01	0.01	3.6	1.7	1.9
Apple	Chlorfenapyr	3,653,855 x + 8117	0.9999	0.97	0.01	4.6	2.6	2.7
	Tralopyril	2,400,056 x + 13,172	0.9992	0.99	0.01	3.1	1.5	2
Tea	Chlorfenapyr	3,486,223 x − 7105	0.9991	0.93	0.01	6.2	7	2.2
	Tralopyril	2,395,137 x + 9500	0.9996	0.99	0.01	4.1	1.7	2.6

Note: ME: matrix/ solvent; LOQ: limit of quantification. “*n* = 5” represents that each treatment was repeated five times. “20*” means that the level of 20 mg kg^−1^ was only added in tea samples.

**Table 2 foods-11-01246-t002:** Residue and limit evaluation of chlorfenapyr and tralopyril bromide in market monitoring samples.

Sampling Locations	Market Type	Matrix	Chlorfenapyr		Tralopyril	
			Detection Quantity (mg kg^−1^)	Exceed Standard	Detection Quantity (mg kg^−1^)	Exceed Standard
Yabo RT-Mart	Supermarket	Chinese cabbage	0.112	No	—	—
		cowpea	0.404	unset limits	0.01	unset limits
		leek	0.022	unset limits	—	—
Tesco Lifestyle	Minimarket	beans	0.048	unset limits	—	—
		eggplant	0.099	No	—	—
		leek	0.224	unset limits	—	—
RT-Mart	Supermarket	eggplant	0.192	No	—	—
		cowpea	0.019	unset limits	—	—
New Age Supermarket	Supermarket	beans	0.042	unset limits	—	—
		cucumber	0.225	No	—	—
Jiayue Supermarket	Supermarket	cabbage	0.104	No	—	—
		leek	0.054	unset limits	—	—
Four Seasons Supermarket	Minimarket	cabbage mustard	0.023	No	—	—
		leek	0.262	unset limits	—	—
		pak choi	0.281	No	—	—
		tea	0.111	unset limits	—	—
Provincial Farmer’s Market	Farmer’s market	cucumber	0.017	No	—	—
		cowpea	0.024	unset limits	—	—
		tea	0.021	No	—	—
Wide Supermarket	Minimarket	eggplant	0.050	No	—	—
		cabbage mustard	0.014	No	—	—
		pak choi	0.014	No	—	—
Lattice Supermarket	Supermarket	cowpea	0.172	unset limits	—	—
		cabbage mustard	0.462	Yes	—	—
		pak choi	0.025	No	—	—
Wuma Farmer’s Wholesale Market	Farmer’s market	cucumber	0.246	No	—	—
		tea	0.024	unset limits	—	—
Yingsheng Farmer’s Market	Farmer’s market	leek	0.056	unset limits	—	—
C ‘mon Lotus Supermarket	Supermarket	tea	0.092	unset limits	—	—
Zaohang Farmer’s Wholesale Market	Farmer’s market	eggplant	0.027	No	—	—
		cowpea	0.024	unset limits	—	—
Rundu Supermarket	Minimarket	beans	0.036	unset limits	—	—
		cabbage	0.015	No	—	—
		cucumber	0.044	unset limits	—	—
		leek	0.026	unset limits	—	—
		tea	0.019	No	—	—
Baibayu Convenient Market	Farmer’s market	beans	0.041	unset limits	—	—
		leek	0.182	unset limits	—	—

Note: “—” means not detected. “Exceed standard” means the detected residues had a higher MRL (maximum residue limit). “Unset limits” means the maximum residue limit is not set.

**Table 3 foods-11-01246-t003:** Final residues of chlorfenapyr in cabbage.

Test Site	Application Dose (g ha^−1^)	Application Times (Freq)	Harvest Interval (Days)	Residual Quantity (mg kg^−1^)	STMR (mg kg^−1^)	HR (mg kg^−1^)
Liaoning, Shanxi, Beijing, Shandong, Shanghai, Anhui, Hunan, Jiangxi, Guangxi, Chongqing, Guizhou, Guangdong	120	2	14	<0.01, 0.015, 0.016, 0.067, 0.070, 0.089, 0.121, 0.210, 0.213, 0.310, 0.337, 0.471	0.105	0.471
			21	<0.01 (8), 0.014, 0.031, 0.046, 0.144	0.01	0.144

Note: STMR: supervised trials median residue; HR: the highest residue.

**Table 4 foods-11-01246-t004:** Calculation table for dietary risk assessment of chlorfenapyr and tralopyril.

Pesticide	Food Types	Intake (kg)	Residue (mg kg^−1^)	Sources of Residues	NEDI (mg)	Daily Intake Allowed (mg)	Risk Probability
Chlorfenapyr (no consideration of tralopyril)	Dark vegetables	0.0915	10	Residue limit (pak choi)	0.915	ADI × 63	NEDI/(ADI × 63)
	Light vegetables	0.1837	0.404	Market monitoring (cowpea)	0.074		
	Fruits	0.0457	0.54	Residue limit (mulberry)	0.025		
	Salt	0.012	20	Residue limit (tea)	0.240		
	Soy sauce	0.009	0.111	Market monitoring (ginger)	0.001		
	Sum	1.0286			1.255	1.929	65.1 %
Chlorfenapyr (consideration of tralopyril)	Food types	Intake (kg)	Residue (mg kg^−1^)	Sources of residues	NEDI (mg)	Daily intake allowed (mg)	Risk probability
	Dark vegetables	0.0915	10	Residue limit (pak choi)	0.915	ADI × 63	NEDI/(ADI × 63)
	Light vegetables	0.1837	0.504	Market monitoring (cowpea)	0.093		
	Fruits	0.0457	0.54	Residue limit (mulberry)	0.025		
	Salt	0.012	20	Residue limit (tea)	0.240		
	Soy sauce	0.009	0.111	Market monitoring (ginger)	0.001		
	Sum	1.0286			1.274	1.929	66.0%

Note: NEDI: national estimated daily intake; ADI: acceptable daily intake.

## Data Availability

Data is contained within the article and Appendix A.

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
