# Peer review of "Determination of Market, Field Samples, and Dietary Risk Assessment of Chlorfenapyr and Tralopyril in 16 Crops"

_foods, 2022, doi:10.3390/foods11091246_

Round 1
Reviewer 1 Report
The manuscript need to be revised and some corrections are required.
- Title: is not good. Too large. The title does not attract the public attention.
- Introduction: More information about the toxicity of the chlorfenapyr is necessary.
- Material and methods: the brand and origin of the reagents and other instrumental used is necessary.
- Results & Discussion: the intake of chlorfenapyr is not correct. What is about the future concerns?
Author Response
response to reviewer 1
Response: thank you for your suggestion of our work. Your comments were highly insightful and enabled us to greatly improve the quality of our manuscript. We have carefully revised it based on your suggestions “point to point” Thank you again for your valuable and helpful suggestions.
- Title: is not good. Too large. The title does not attract the public attention.
Response: thank you for your suggestion. The title “Market sample monitoring, terminal residue, dissipation be-havior and dietary risk assessment of chlorfenapyr and tra-lopyril in 16 crops” has been modified into “Determination of market, field samples and dietary risk as-sessment of chlorfenapyr and tralopyril in 16 crops”. If there is a problem, we expect guidance and advice of expert.
- Introduction: More information about the toxicity of thechlorfenapyr is necessary.
Response: thank you for your suggestion. We have added more information about the toxicity of the chlorfenapyr. The content was shown in 1Introduction part as follows: “Chlorfenapyr is a highly concentrated pesticide, which can be toxic to humans, birds, fish and silkworms, can also damage the DNA of liver, spleen and kidney cells of mice [2,3]. The information has been added and marked red in the revised manuscript.
- Material and methods: the brand and origin of the reagents and other instrumental used is necessary.
Response: thank you for your suggestion. We have added more information about the brand and origin of the reagents as follows: Reference standards of chlorfenapyr (98.1%) and tralopyril (99.3%) were purchased from China Standard Substance Center (Germany Dr, Beijing, China). High-performance liquid chromatography (HPLC)-grade acetonitrile and dichloromethane were purchased from Merck (Thermo Scientific, Darmstadt, Germany). Analytical grade sodium chloride (NaCl) and anhydrous magnesium sulfate (MgSO4) were purchased from Sinopharm (Thermo scientific, Shanghai, China). The purification agents of GCB, PSA, and C18 were purchased from Angela Technologies Co., Ltd. (Agela, Tianjin, China). The 240 g L-1 chlorfenapyr SC (suspension concentrate) was purchased from Shandong United Pesti-cide Industry Co., Ltd. (Rennes, Jinan, China). The information has been added and marked red in the revised manuscript of Material and methods part.
- Results & Discussion: the intake of chlorfenapyr is not correct. What is about the future concerns?
Response:1) thank you for your suggestion. I am sorry to say that we editor incorrect data of residue in light vegetables, so we have changed the data “0.404” into “0.504” under the consideration of tralopyril, but the final risk values were correct due to we calculate it via collect data of 0.504. The information has been added and marked red in the revised manuscript of Material and methods part. Thank you again for your valuable and helpful suggestions.
|
Pesticide |
Food types |
Intake (kg) |
Residue (mg kg-1) |
Sources of residues |
NEDI (mg) |
Daily intake allowed (mg) |
Risk probability |
|
Chlorfenapyr (no consideration of tralopyril) |
Dark vegetables |
0.0915 |
10 |
Residue limit (pakchoi) |
0.915 |
ADI×63 |
NEDI/(ADI×63) |
|
Light vegetables |
0.1837 |
0.404 |
Market monitoring (cowpea) |
0.074 |
|||
|
Fruits |
0.0457 |
0.54 |
Residue limit (mulberry) |
0.025 |
|||
|
Salt |
0.012 |
20 |
Residue limit (tea) |
0.240 |
|||
|
Soy sauce |
0.009 |
0.111 |
Market monitoring (ginger) |
0.001 |
|||
|
Sum |
1.0286 |
|
|
1.255 |
1.929 |
65.1 % |
|
|
Chlorfenapyr (consideration of tralopyril) |
Food types |
Intake (kg) |
Residue (mg kg-1) |
Sources of residues |
NEDI (mg) |
Daily intake allowed (mg) |
Risk probability |
|
Dark vegetables |
0.0915 |
10 |
Residue limit (pakchoi) |
0.915 |
ADI×63 |
NEDI/(ADI×63) |
|
|
Light vegetables |
0.1837 |
0.504 |
Market monitoring (cowpea) |
0.093 |
|||
|
Fruits |
0.0457 |
0.54 |
Residue limit (mulberry) |
0.025 |
|||
|
Salt |
0.012 |
20 |
Residue limit (tea) |
0.240 |
|||
|
Soy sauce |
0.009 |
0.111 |
Market monitoring (ginger) |
0.001 |
|||
|
Sum |
1.0286 |
|
|
1.274 |
1.929 |
66.0% |
2) The amount of agrochemicals application changes every year, and the application amount is large. However, long-term usage causes pesticide accumulation and residue, and vegetables and fruits are daily consumables. Therefore, timely risk assessment can ensure food safety problems of humans. In brief, this research could guide the rational use of chlorfenapyr and tralopyril and serve as a reference for the establishment of an MRL in China. Furthermore, conducting risk assessment of pesticide residues contributes to food safety risk management, risk communication and consumer health (the content marked red was shown in 4 conclusion part).
On the other hand, in the present research, the dietary risk assessment of chlorfenapyr is evaluated with and without tralopyril, because the metabolite tralopyril has high toxicity in comparison to the parent chlorfenapyr and probably cause greater food safety problems. Both chlorfenapyr and tralopyril were assessed more comprehensively than other studies can only consider food risk of chlorfenapyr, which can more valuably reflect the food safety to heathy risk of humans (the content marked red was shown in 3.7 Dietary risk assessment part).
Furthermore, the risk values were 65.1 and 66.0%, respectively. Although values were greatly lower than RQ = 100%, which indicated no risk, the values were at a high level. The results remind the government to strengthen the management of agrochemicals application. Therefore, food risk assessment is necessary. The above is our response to your question of “What is about the future concerns?”. Thank you again for your valuable and helpful suggestions.
Reviewer 2 Report
Regarding the manuscript "Market sample monitoring, terminal residue, dissipation behavior and dietary risk assessment of chlorfenapyr and tralopyril in 16 crops" in judge it contains technical results being presented as a report. The manuscript is within the scope of analytical chemistry journals, firstly.
The grade of novelty is moderate to weak. However, I would like to suggest the following improvements:
- L49: Tea with lowercase.
- L53: remove "for 16". It is quite specific.
- L104: here and in the whole text, the unit is g ha-1, not g a.i. ha -1.
- L123 and 128 and so on: Is g in uppercase I think.
- L137: please standardize the units mL/min or mL min-1 in the whole text.
- L182 and so on: remove bw from the unit.
- L183: median word is unformatted.
- L188: avoid contractions.
- Figure 1: letters inside the figure are unreadable.
- Figure 2: check the y axis captions. I have not understood the differences between colors and circles sizes as both of them are "recoveries".
- Table 2 seems unformatted.
- Figure 3: please inform the RMSE besides the R2.
- Figure 3: here you quote day as "d". In text as "day", standardize it.
I have not able to check self citations and plagiarism.
Author Response
response to reviewer 2
Regarding the manuscript "Market sample monitoring, terminal residue, dissipation behavior and dietary risk assessment of chlorfenapyr and tralopyril in 16 crops" in judge it contains technical results being presented as a report. The manuscript is within the scope of analytical chemistry journals, firstly.
The grade of novelty is moderate to weak. However, I would like to suggest the following improvements:
Response: thank you for your suggestion of our work. Your comments were highly insightful and enabled us to greatly improve the quality of our manuscript. We have carefully revised it based on your suggestions “point to point” Thank you again for your valuable and helpful suggestions.
- L49: Tea with lowercase.
Response: thank you for your suggestion of our work. The “Tea” has been modified into “tea” in L51 of modification manuscript.
- L53: remove "for 16". It is quite specific.
Response: thank you for your suggestion of our work. The “for 16” has been removed in L59 of modification manuscript.
- L104: here and in the whole text, the unit is g ha-1, not g a.i. ha -1.
- Response: thank you for your suggestion of our work. All “g a.i. ha-1” have been modified g ha-1 in modification manuscript.
- L123 and 128 and so on: Is g in uppercase I think.
Response: thank you for your suggestion of our work. After our investigation and inspection, the uppercase and lower case are both permitted, and we verified the previous paper in this food journal use the lower case (shown in the following picture), so we keep on the lower case. Thank you again for your valuable and helpful suggestions and we recognize more forms about units
- L137: please standardize the units mL/min or mL min-1in the whole text.
- Response: thank you for your suggestion of our work. All “mL/min” have been modified into “mL min-1” in modification manuscript.
- L182 and so on: remove bw from the unit.
- Response: thank you for your suggestion of our work. All “bw” have been removed in modification manuscript.
- L183: median word is unformatted.
- Response: thank you for your suggestion of our work. The median word has been formatted in L198 of modification manuscript.
- L188: avoid contractions.
- Response: thank you for your suggestion of our work. “The RQ represents the risk quotient. When the RQ ≤ 100%, it indicated acceptable risk, otherwise it couldn’t” has been modified “The RQ represents the risk quotient. When the RQ ≤ 100%, it indicated acceptable risk, otherwise it could not” The modification has been shown in L203 of modification manuscript.
- Figure 1: letters inside the figure are unreadable.
- Response: thank you for your suggestion of our work. Letters (A and B) inside the figure are marked in Figure 1.
- Figure 2: check the y axis captions. I have not understood the differences between colors and circles sizes as both of them are "recoveries".
Response: thank you for your suggestion of our work. The bubble diagram was made via the imageGP system to show the recoveries of different crops at three addition levels. The colors and circles sizes are used to all show the size of recoveries, and this form made the result richer and more dynamic. The y axis captions show different addition levels of chlorfenapyr (A) or (B). Furthermore, we added the note under the figure 2. If you have further guidance, we will make better modifications according to your suggestions later. Thank you again for your valuable and helpful suggestions.
Note: The colors and circles sizes both show the amount of recovery.
- Table 2 seems unformatted.
Response: thank you for your suggestion of our work. Table 2 has been formatted.
- Figure 3: please inform the RMSE besides the R2.
- Response: thank you for your suggestion of our work. The RMSE has been added in Figure 3 which was shown in modification manuscript.
- Figure 3: here you quote day as "d". In text as "day", standardize it.
- Response: thank you for your suggestion of our work. The "d" in Figure 3 has been modified into "day".
- I have not able to check self-citations and plagiarism.
- Thank you for your suggestion. The manuscript will be sent to reviewers when there is no plagiarism checked by journal of Therefore, the manuscript has no self-citations and plagiarism.
Reviewer 3 Report
The authors reported a study to determine chlorfenapyr and tralopyril residues in registered crops of China by using a sensitive and straightforward method. They also investigate the terminal residues and dissipation dynamics on cabbage from various regions in China. The authors should address the following issues:
- Line 36, please provide more recent production value than 2017.
- Most of the section “3.1 HPLC–MS/MS optimization conditions” needs to move to the materials and methods section.
- Table 1, 3, and 4 are missing the footnotes.
- What are the findings in the literature? Authors should compare their values with the literature.
- What is the novelty of this research? Authors need to emphasize the novelty of their research.
Author Response
response to reviewer 3
- The authors reported a study to determine chlorfenapyr and tralopyril residues in registered crops of China by using a sensitive and straightforward method. They also investigate the terminal residues and dissipation dynamics on cabbage from various regions in China. The authors should address the following issues:
Response: thank you for your suggestion of our work. Your comments were highly insightful and enabled us to greatly improve the quality of our manuscript. We have carefully revised it based on your suggestions “point to point”. Thank you again for your valuable and helpful suggestions.
- Line 36, please provide more recent production value than 2017.
- Response: thank you for your suggestion. More recent production value was modified in Line 38 of the modification manuscript.
- Most of the section “3.1 HPLC–MS/MS optimization conditions” needs to move to the materials and methods section.
Response: thank you for your suggestion. Most of the section “3.1 HPLC–MS/MS and GC- MS/MS optimization conditions” have been moved to the materials and methods section. The modified information has been added and marked red in the revised manuscript of Material and methods part. Thank you again for your valuable and helpful suggestions.
- Table 1, 3, and 4 are missing the footnotes.
Response: thank you for your suggestion. The footnotes of Table 1, 3, and 4 were all added. The modified information has been added and marked red in the revised manuscript. Thank you again for your valuable and helpful suggestions.
- What are the findings in the literature? Authors should compare their values with the literature.
- Response: thank you for your suggestion. We discuss their values with the previous literature, which shown in follows:
- The dietary risk assessment of chlorfenapyr is evaluated with and without tralopyril, because the metabolite tralopyril has high toxicity in comparison to the parent chlorfenapyr and probably cause greater food safety problems. Both chlorfenapyr and tralopyril were assessed more comprehensively than other studies which can only consider food risk of chlorfenapyr. Therefore, the present study can more valuably reflect the food safety to heathy risk of humans. The content was shown in 3.7. Dietary risk assessment part of the modification manuscript.
- What is the novelty of this research? Authors need to emphasize the novelty of their research.
- The amount of agrochemicals application changes every year, and the application amount is large. However, long-term usage causes pesticide accumulation and residue, and vegetables and fruits are daily consumables. Therefore, timely risk assessment can ensure food safety problems of humans. In brief, this research could guide the rational use of chlorfenapyr and tralopyril and serve as a reference for the establishment of an MRL in China. Furthermore, conducting risk assessment of pesticide residues contributes to food safety risk management, risk communication and consumer health (the content marked red was shown in 4 conclusion part).
- On the other hand, in the present research, the dietary risk assessment of chlorfenapyr is evaluated with and without tralopyril, because the metabolite tralopyril has high toxicity in comparison to the parent chlorfenapyr and probably cause greater food safety problems. Both chlorfenapyr and tralopyril were assessed more comprehensively than other studies can only consider food risk of chlorfenapyr, which can more valuably reflect the food safety to heathy risk of humans (the content marked red was shown in 3.7 Dietary risk assessment part). However, less literature was reported the risk co-chlorfenapyr and tralopyril in multiful crops. Furthermore, the risk values were 65.1 and 66.0%, respectively. Although values were greatly lower than RQ = 100%, which indicated no risk, the values were at a high level. The results remind the government to strengthen the management of agrochemicals application. Therefore, food risk assessment is necessary. Thank you again for your valuable and helpful suggestions. We hope to get your approval.